# Calibration of the DSCOVR EPIC visible and NIR channels using MODIS Terra and Aqua data and EPIC lunar observations

Igor V. Geogdzhayev[1], Alexander Marshak[2]

[1]Department of Applied Physics and Applied Mathematics, Columbia University/ NASA Goddard Institute for Space Studies, New York, NY 10025, USA
[2]NASA Goddard Space Flight Center, Greenbelt, MD, 20771

*Correspondence to*: Igor Geogdzhayev (igor.v.geogdzhayev@nasa.gov)

**Abstract.** The unique position of the Deep Space Climate Observatory (DSCOVR) Earth Polychromatic Imaging Camera (EPIC) at the Lagrange 1 point makes an important addition to the data from currently operating low orbit Earth observing instruments. EPIC instrument does not have an onboard calibration facility. One approach to its calibration is to compare EPIC observations to the measurements from polar orbiting radiometers. Moderate Resolution Imaging Spectroradiometer (MODIS) is a natural choice for such comparison due to its well-established calibration record and wide use in remote sensing. We use MODIS Aqua and Terra L1B 1km reflectances to infer calibration coefficients for four EPIC visible and NIR channels: 443 nm, 551 nm, 680 nm and 780 nm. MODIS and EPIC measurements made between June 2015 and June 2016 are employed for comparison. We first identify favorable MODIS pixels with scattering angle matching temporarily collocated EPIC observations. Each EPIC pixel is then spatially collocated to a subset of the favorable MODIS pixels within 25 km radius. Standard deviation of the selected MODIS pixels as well as of the adjacent EPIC pixels is used to find the most homogeneous scenes. These scenes are then used to determine calibration coefficients using a linear regression between EPIC counts/sec and reflectances in the close MODIS spectral channels. We present thus inferred EPIC calibration coefficients and discuss sources of uncertainties. The Lunar EPIC observations are used to calibrate EPIC O2 absorbing channels (688 nm and 764 nm) assuming that there is a small difference between moon reflectances separated by ~10 nm in wavelength provided the calibration factors of the red (680 nm) and near-IR (780 nm) are known from comparison between EPIC and MODIS.

## 1 Introduction

The Deep Space Climate Observatory (DSCOVR) occupies a unique location among Earth-observing instruments in the Lagrange point L1 between the Sun and the Earth at about 1.5 million kilometers from Earth. The spacecraft actively maintains itself in a Lissajous orbit around L1. This position allows DSCOVR Earth Polychromatic Imaging Camera (EPIC) to view the entire sunlit Earth's hemisphere (Fig. 1, right panel). Since the launch in June 2015 EPIC provided regular Earth images in ten narrow spectral channels ranging from UV to near IR. The Earth-observing geometry of the EPIC instrument is characterized by nearly constant scattering angle between 168.5° and 175.5° and the distance from Earth between 1.4 and 1.6 million kilometers (Fig. 2). In that EPIC's viewing geometry differs significantly from instruments on sun-synchronous orbits which

rarely view Earth at such large scattering angles. For comparison, depending on the season, latitude and scan view angle, the scattering angle for MODIS is typically in a wide range between 110° and 175°. The Suomi-NPP VIIRS instrument, due to its wider range, covers even larger range of angle including the whole backscattering region. The almost back scattering EPIC observations are a direct consequence of its position at L1.

The large scattering angle of EPIC observations is a significant difference compared to the observations from low Earth orbit (LEO) instruments. The large scattering angles may present challenges for some retrievals. However they may also be desirable for other applications. For example, the position of the water surface glint in the center of the sunlit hemisphere allows better coverage where LEO instruments often see glint. Also of note is the lack of shadows for vertically extended scenes.

Measurements in the backscattering region allow observations and characterizations of the glint caused by oriented ice crystals in clouds (Marshak et al., 2017b). Availability of these measurements also allows better vegetation monitoring (Marshak and Knyazikhin, 2017). Using the back scattering radiation it is possible to estimate the total Leaf Area Index and its sunlit portion separately (Yang et al., 2017); this is important because direct and diffusely illuminated leaves have different photosynthetic rates. Thanks to its position and viewing geometry, the EPIC instrument offers an improved temporal sampling compared to

instruments on the sun-synchronous orbit. It samples the entire sunlit hemisphere 10-20 times per day. Compared to other instruments on a geostationary orbit, EPIC provides improved coverage in high latitudes hemispheres. It thus has the potential to augment remote sensing observations in such applications as aerosol, cloud, sulphur dioxide and ozone amounts as well as vegetation properties (Marshak et al., 2017a). EPIC data are used for the remote sensing of clouds (Yang et al., 2013) and dust plumes with oxygen A and B bands (Xu et al., 2017); it also provides multi-spectral UV SO2 measurements of the sunlit Earth

disk (Carn et al., 2016). EPIC observations are also used to measure ozone, cloud reflectivity, and erythemal irradiance (Herman et al., 2017).

Radiometric calibration of the measurements is a required first step for many of the above applications. The EPIC instrument does not have in-flight calibration capabilities making determining the calibration coefficients and monitoring their stability

by means of vicarious calibration efforts a necessity. One approach to its calibration is to compare EPIC observations to the measurements from polar orbiting radiometers. Another is to use the images of the Moon regularly observed by the instrument.

Haney et al. (2016) investigated the calibration of EPIC Version 1 data using MODIS and VIIRS using the data aggregated on a .5x.5 degree grid with matching viewing geometry. They found that a navigation correction reduced the uncertainties in the

calibration gain. The straylight correction was found to reduce the fit offsets and gains for all considered channel pairs. Yu and Wu (2016) investigated the inter-calibration between Advanced Himawari Imager (AHI) in a geostationary orbit and VIIRS. They found strong linear relationship between the paired bands. The radiometric calibration between the two instruments was shown to agree within 5%.

In this study we use Moderate Resolution Imaging Spectroradiometer (MODIS, King et al., 2003), Level 1b reflectances and collocated EPIC measurements to derive the calibration coefficients in four EPIC visible and near IR (NIR) channels. EPIC UV channels (317.5, 325, 340 and 388 nm) are calibrated using LEO instruments Aura/OMI and Suomi-NPP/OMPS (Herman et al., 2017). We derive calibration gains for the initial (Version 1) and Version 2 releases of the EPIC data base on all available

contemporaneous MODIS Aqua and Terra data. The key difference between Version 1 and Version 2 data is the applied straylight correction together with flat-fielding that made bright pixels brighter and dark pixels darker (Marshak et al., 2017a). MODIS data are a natural choice for such comparison due to their well-established calibration record and wide use in the remote sensing applications.

## 2 Data

EPIC L1B data were obtained from NASA Langley Atmospheric Science Data Center (ASDC). The EPIC sampling size at nadir (at the center of the image) is about 8x8 km$^2$ (it is 10x10 km$^2$ when the EPIC point spread function is applied) and increases towards the edges. The radiometric resolution of EPIC data is 12 bits per pixel. To reduce the amount of data transmitted from DSCOVR, for all but blue channel (443 nm) four pixels are averaged on-board the spacecraft resulting in the effective spatial resolution at nadir of approximately 18 km.

We use MODIS Aqua and Terra L1B Collection 6 1-km reflectances obtained from the Level-1 and Atmosphere Archive & Distribution System (LAADS) Distributed Active Archive Center (DAAC). Note that the MODIS reflectance, as well as EPIC, is the true reflectance multiplied by the solar zenith angle (MODIS Level 1B Product User's Guide, 2006). We will refer to this quantity as simply "reflectance". MODIS data have radiometric resolution of 12 bits per pixel and the calibration design

requirements of 2% for reflectance and 5% for radiance in the solar bands (Toller et al. 2013). MODIS channels number 3, 4, 1, 2 are matched with for four EPIC visible and NIR channels: 443 nm, 551 nm, 680 nm and 780 nm, respectively, as shown in Table 1. Figure 3 shows normalized filter functions for the corresponding channels. As one can see from the figure EPIC channels are significantly narrower compared to MODIS. The best spectral match is for the overlapping green channels, while the largest spectral difference of about 80 nm is observed between the NIR channels of the two instruments.


Two versions of EPIC data are used in this study. The initial release that covers the period between June 2015 and September 2016 will be referred as Version 1. The data from the second release (Version 2) cover the period from June 2015 and March 2017. Version 2 data include refined geolocation, flat-fielding and straylight correction algorithm (Marshak et al., 2017a). These differences require the derivation of separate sets of calibration coefficients for each of the two versions. This is

discussed in more details in the "Straylight correction" section below.

| EPIC channel (Full Width in nm) | MODIS Band (Bandwidth) |
|---|---|
| 443±1 nm (3±0.6) | 3 (459-479nm) |

| | |
|---|---|
| 551±1 nm (3±0.6) | 4 (545-565nm) |
| 680±0.2 nm (3±0.6) | 1 (620-670nm) |
| 779.5±0.3 nm (2±0.4) | 2 (841-876nm) |

**Table 1**. EPIC-MODIS channel correspondence. (For simplicity, for the rest of the paper we will call the EPIC NIR channel 780 nm).

## 3 Analysis

To derive EPIC calibration coefficients we first identify favorable MODIS pixels. For each EPIC image we find MODIS
pixels that match the EPIC scattering angle to within 0.5° and are temporarily collocated to within 10 min. EPIC exposure time varies between 0.046 second for the 443 nm channel to 0.025 second for the 780 nm channel. During these times the Earth rotates less than 20 meters. Compared to many low-orbit radiometers that perform cross-track scans to create an image, EPIC filter-wheel design means that image acquisition is nearly instantaneous. There is a time lag in the data (Marshak et al., 2017b) acquisition between different EPIC spectral channels associated with the rotation of the filter wheels: ~3 min difference
between blue (443 nm) and green (551 nm), ~4 min between blue and red (680 nm). In 4 min a point on the equator will rotate 1° or approximately 110 kilometers. The geolocation algorithm ensures the spatial collocation of the spectral channels for each pixel. Because of the time lag the temporal collocation is done separately for each spectral channel. We limit the solar zenith angle (SZA) of both EPIC and MODIS pixels to be no more than 60° to avoid scenes with low illumination and scenes where the curvature of the Earth may play a significant role potentially complicating the comparison. Depending on the season such
favorable pixels typically occur between 30S and 30N. We ignore the EPIC pixels that are affected by specular component (Marshak et al., 2017b). We then select EPIC pixels that have a minimum of 40 MODIS pixels within 25 km radius. Relative standard deviation (defined as the ratio of the absolute standard deviation to the mean) is then calculated for the matching MODIS and EPIC pixels. In the latter case 5x5 pixel neighborhood are used to calculate the standard deviation. The value of the relative standard deviation is used to select the most homogeneous scenes.

To determine the calibration coefficients two independent methods are used: the first is based on calculating the linear regression between EPIC counts/sec and MODIS reflectances in the closest MODIS spectral channel for the most homogeneous scenes. The second is based on finding the mean MODIS/EPIC (M/E) ratio for bright (MODIS reflectance is greater than 0.6) homogeneous scenes.

To calculate the linear regressions we use the most uniform collocated scenes with the relative standard deviations smaller than a threshold value. Because of the time delays in data acquisition between different EPIC channels and occasional gaps in data transmission, the number of matching MODIS pixels may differ for each regression. The threshold standard deviation is selected separately for each channel. It is reduced until the correlation coefficient of the resulting regression stops increasing

or the number of matching points falls below a certain value. The resulting regressions for Version 2 data are shown in Figure 4. The relative standard deviation of MODIS and EPIC points included in the regressions was between 0.5% and 1% depending on the channel. In this approach the spatial homogeneity threshold was the greatest limiting factor to the number of EPIC and MODIS pairs.

The second approach to deriving the regression coefficients that we employed is based on calculating the ratio of MODIS reflectance to EPIC count for all available matching scenes with high reflectance and relative standard deviation less than 10%. We then select the pixels for which the MODIS reflectance is greater than 0.6. Such pixels represent between 10% and 15% of the total. Note that the reflectance threshold is only applied to the matching MODIS pixels, which are assumed to be well calibrated and stable. These scenes are binned according to the relative standard deviation of the MODIS reflectance and the mean M/E ratio is calculated for each bin. The mean bin values are then extrapolated to the ideal case of a completely uniform scene (zero standard deviation) using a linear regression. The extrapolated value is then taken to be the calibration coefficient. Because EPIC observations are made in the backscattering region the sunglint usually occurs in the center of the image. Bright sunglint can exceed 0.6 reflectance however such scenes are not spatially homogeneous and are screened out by the relative standard deviation requirement.

This approach assumes zero intercept value. Because the number of points contributing to the M/E ratio calculations at least three orders of magnitude greater than the number of points selected for the regression method the two may be considered to be essentially independent. The four panels of Figure 5 illustrate the M/E analysis for the four EPIC channels. The squares and whiskers show respectively the mean and the standard variation of the ratios in each bin. Straight lines show linear regressions. One can see that for the relative standard deviation below 10% the mean M/E values are similar for every bin, so that the extrapolated value does not differ from the mean by more than 1%. The differences in the gain coefficients calculated using the two methods is given in Table 2 which shows the officially published gain coefficients for the two dataset versions. These coefficients are also publicly available at https://eosweb.larc.nasa.gov/project/dscovr/DSCOVR_EPIC_Calibration_Factors_V02.pdf. When comparing the gain coefficients from the two methods below we do not force the regression through zero. Doing so would reduce the independence of the two approaches, as it would effectively ignore the contribution of the dark scenes to the regression.

For Version 1 data the differences between the two calibration methods are between approximately 3% for the 443 nm channel and less than 0.5% for the 780 nm channel. An improved agreement is observed for the Version 2 data, for which the differences range between 0.1% and 1.4 %. A dependence of the M/E ratio on the relative standard deviation of the MODIS pixels may potentially exist because of the different effect of the scene's cloud or surface inhomogeneity on the two instruments due to different viewing geometry. However, this approach does not assume its existence, as the gain coefficients are obtained from

the extrapolation to the "ideal" case of completely uniform scene thus accounting for any potential systematic behavior. If the relation to the standard deviation is completely random, the resulting coefficients will be similar to what one would obtain by simply calculating the mean M/E ratio. ==An improved agreement between the two methods for Version 2 data may partially be attributed to more accurate geolocation algorithm (Marshak et al., 2017a). Other factors such as the straylight correction and spectral correction are discussed in the following sections.==

| Version 1 | | |
|---|---|---|
| EPIC Channel | Calibration coefficients published on July 16, 2016 | M/E Ratio/regression difference (%) |
| 443 nm | 8.80E-6 | 2.79 |
| 551 nm | 6.90E-6 | 1.98 |
| 680 nm | 1.00E-5 | 1.01 |
| 780 nm | 1.50E-5 | 0.41 |
| Version 2 | | |
| EPIC Channel | Calibration coefficients published on July 6, 2017 | M/E Ratio/regression difference (%) |
| 443 nm | 8.34E-06 | 0.1 |
| 551 nm | 6.66E-06 | 0.5 |
| 680 nm | 9.30E-06 | 0.5 |
| 780 nm | 1.435E-05 | 1.4 |

**Table 2.** Ratio/regression gain coefficients differences of the gain coefficients for the four EPIC channels.

## 4 Straylight correction effect

Compared to the initial release of the EPIC data (Version 1) the second release (Version 2) includes a number of changes such as refined data geolocation for each filter and flat-fielding (correcting for CCD irregularities). In addition, a straylight correction algorithm (based on laboratory measurements and in-flight lunar observations) is used. Straylight refers to the illumination of multiple CCD pixels by a point light source. The main causes of straylight are the diffraction and ghosting (reflections between the CCD and filter surfaces). The magnitude of the effect is different for each spectral channels and is stronger in the visible compared to the UV primarily because of the larger dynamic range in the visible. The straylight correction has a two-fold effect on the EPIC counts: it decreases the count value for dark scenes and increases it for bright scenes within an EPIC image as the total radiant energy must remain constant. These changes results in a smaller intercept and gain of the linear regression compared to the case with no straylight correction. A schematic illustration of the effect of the straylight correction is shown in Figure 6.

Version 1 data (no straylight correction) and Version 2 data which includes straylight corrected data can be used to evaluate the effect. The regression analysis and the M/E ratio method described above were applied to both sets. The analysis shows a 4-9% reduction in the gain coefficients for the new data compared to the initial release. Consistent with expectations the reduction is smaller for the blue and green channels compared to the red and NIR ones. This reduction is observed for gain coefficients calculated by the two methods and is consistent with the expected effect of the straylight correction. Note that in order to make the two sets comparable, the spectral correction (see next section) was not applied to the Version 2 data.

Similarly, we compared the regression offsets for the two datasets. Absolute reductions of the offset coefficients are observed for all channels, consistent with the expectations. The reductions range to between 1.5 and 3.7 times illustrated in Figure 7.

## 5 Spectral correction

The differences in the position and spectral width of the corresponding EPIC and MODIS channels may result in discrepancies when scenes with different spectral signatures are observed by the two instruments (Chander, 2013). In version 2 calibration to compensate for these differences we employed spectral band adjustment factors (SBAFs) which convert MODIS reflectance values to equivalent EPIC reflectance for various surface types. These factors in the form of linear regression coefficients were obtained from https://cloudsgate2.larc.nasa.gov/cgi-bin/site/showdoc?mnemonic=SBAF; they are based on the analysis of the SCIAMACHY hyperspectral data for various surface targets to account for the differences in MODIS and EPIC spectral response functions (Scarino et al., 2016).

In addition, the minimum and maximum reflectance values were identified based on the same source. MODIS pixels were spectrally adjusted if the reflectance was within these limits using the SBAFs for the appropriate land cover type. Deep convection clouds spectral corrections were applied to scenes with reflectance higher than 0.6. To identify the land cover type for each matching EPIC pixel we use a data set developed by Channan et al (2014). The dataset is a .5x.5 degree reprojected version of the Global Mosaics of the standard MODIS land cover type data product (MCD12Q1) in the IGBP Land Cover Type Classification. Separate adjustment factors were used for MODIS Aqua and MODIS Terra data. One may assume that bright pixels mostly represent cloudy scenes, dark pixels - water, and intermediate values represent deserts. However, it would be impossible to classify the pixels with certainty within the framework of the crude approach, where SBAFS for clouds are assumed for bright scenes with reflectance greater 0.6 and a fixed surface classification map is used. This represents a limitation of the current approach and may be responsible for some spread in the regressions.

Table 3 shows the effect of spectral adjustment on calibration gains found using M/E ratio and regression methods. The effect is different for the two methods because of the different scenes used. Bright scenes, assumed to be clouds, were used for the M/E ratio analysis. The regression analysis included dark scenes and scenes of intermediate brightness as well. Overall

accounting for the spectral differences of the matched channels results in changes in the gains coefficients of about 1% .The strongest effect of the spectral correction of 6% in the red channel is due to the larges SBAFs of 10-15% for the scenes of intermediate brightness. Note, that, while spectral correction was not used for the Version 1 data, these scenes were excluded from the analysis and did not have an effect on the Version 1 calibration coefficients. Figure 7 shows the absolute values of the regression offsets for Version 1 and Version 2 data with and without spectral correction. The closeness of the offset values to the ideal case of zero offset can be interpreted as an improvement. One can see that the implementation of the straylight correction in Version 2 reduces the offset values for all channels. In addition spectral correction in Version 2 further reduces offsets thus increasing our confidence in the utility of the spectral correction.

| EPIC channel | Relative difference in gain coefficients , % | |
| --- | --- | --- |
| | M/E ratio analysis | Regression analysis |
| 443 nm | 0.8 | 0.9 |
| 551 nm | 0.1 | 0.5 |
| 680 nm | 1.3 | 6.0 |
| 780 nm | 1.4 | 1.0 |

**Table 3**. The effect of spectral adjustment on calibration gain coefficients for version 2 data. The values in the table are calculated by subtracting the gain coefficients for data with spectral correction from the corresponding coefficients for data without spectral correction and dividing the result by the former.

## 6 Seasonal dependence

The length of the available EPIC dataset allowed us to evaluate the magnitude of any possible temporal change in the derived calibration coefficients. Such a change may potentially be due to two distinct factors: seasonal dependence of the calibration method itself or the degradation of the EPIC instrument. With the data covering only one full seasonal cycle it may be difficult to reveal a seasonal dependence of the calibration procedure and thus separate the two factors. However observing no or small temporal change would be an encouraging sign of both the stability of the instrument and the robustness of the calibration method. To evaluate the temporal changes we calculated M/E ratios separately for three-month periods between September 2015 and February 2017. Only data points with MODIS reflectance greater than 0.6 and EPIC and MODIS relative standard deviation less than 5% are included.

The resulting seasonal dependence is shown on Figure 8. One can see that there is no noticeable trend in the data and the observed differences are within the range of variation of the ratios. The seasonal variability of the gain coefficients calculated from the data in Figure 8 is generally less than 1% (0.9% for 443 nm channel, 0.6 for the 551 nm, 0.4% for 680 nm and 0.6% for 780 nm). We also found that the seasonal variability tends to become smaller for more homogeneous scenes. Thus for

higher standard deviation value of 10% the seasonal variability is approximately 2-2.5% compared to the above values of less than 1% for standard deviation of 5%.

## 7 Calibration of the EPIC O2 absorbing bands using full moon EPIC observations

In general EPIC instrument observes the Moon every two-three months. Several images may be acquired for each observation, each containing several hundred individual Moon pixels.

Moon reflectance $R_\lambda$ increases slowly with wavelength $\lambda$; in most cases (e.g., Ohtake et al., 2010, 2013), a 10 nm difference in wavelength leads to a difference in reflectance in the range of 0.0006-0.0013 or 0.8-1.2%. Based on this, the difference in moon reflectance between the $O_2$ B-band (688 nm) and the 'red' (680 nm) channels as well as between the O2 A-band (764 nm) and the near-IR (780 nm) channels will be within 1.6%. We use the moon reflectance ratios $R_{688}/R_{680} = 1.008$ and $R_{764}/R_{780}$ = 0.984. Since the calibration factors for 680 and 780 nm channels are known from comparisons between EPIC Earth observations and the measurements from polar orbiting radiometers, we can obtain the calibration factors for the O2 absorbing channels at 688 and 764 nm. Indeed, the ratio $F(\lambda_1,\lambda_2)$ of the moon reflectance values measured in counts/sec at two neighboring channels $\lambda_1$ and $\lambda_2$ is very stable (Fig. 9); it is 0.466±0.002 for the 688 over 680 nm ratio and 0.591±0.002 for the 764 over 780 nm one. To avoid the effects of libration the edges of the disk were ignored. Thus, the calibration factor $K$ for 688 nm, can be approximated as

$$K(688) = R_{688}/R_{688}^{counts} = R_{688}/[R_{680}^{counts}\ F(680,688)] =$$
$$= [R_{688}\ K(680)]/[R_{680}\ F(680,688)] = [R_{688}/R_{680}]\ [K(680)/F(680,688)] \approx$$
$$\approx 1.008\ [K(680)/F(680,688)].$$

Similar to 688 nm the calibration factor for 764 nm can be estimated as $K(764) \approx 0.984\ [K(780)/F(780,764)]$. Here $R_\lambda$ and $R_\lambda^{counts}$ are the values of calibrated reflectance and measured counts/sec at wavelength $\lambda$, respectively; $K(\lambda)$ is the multiplicative calibration coefficient expressed as a conversion from counts/sec to reflectance at wavelength $\lambda$ and the ratio $F(\lambda_1,\lambda_2) = R_{\lambda 2}^{counts}/R_{\lambda 1}^{counts}$. Using this technique we obtained the following calibration coefficients: $K(688) = 2.02e-05$ and $K(764)=2.36e-05$.

Note that while both absorbing and non-absorbing channel reflectance change substantially straylight corrections (up to 10%), their ratios $F(\lambda_1,\lambda_2)$ are very stable: the difference between straylight corrected data and with no correction is less than 1.5%. Also note that to calculate the ratios we used both full moon and new moon data separately. The ratios for different moon phases were very close.

**8 Comparison with ROLO-derived calibration coefficients.**

The Robotic Lunar Observatory (ROLO) run by the United States Geological Survey (USGS) provides radiometric calibration and sensor stability monitoring for space-based remote sensing instruments using the Moon as a reference source (Kieffer and Stone, 2005). Using the technique of the minimization of residuals between 16 EPIC moon observations from 2015-08-29, 2016-04-21, 2016-07-19 and 2016-10-14 and the reference ROLO data an independent set of calibration coefficients was developed for all 10 EPIC channels form UV to NIR (Tom Stone, personal communication).

Figure 10 presents the comparison of the six EPIC channels gain coefficients developed in this study (see second column of the Version 2 part of Table 2 and the two gain values for the absorbing channels from the previous section) with the lunar-derived values. The absolute coefficients derived by the two methods agree to within approximately 10%, with ROLO coefficients being systematically lower. In absolute terms the gain coefficients for 4 non-absorbing channels are in a better agreement compared to the two O2 absorbing channels.

The difference with the ROLO coefficients is noticeably greater than the two methods reported in the previous sections and greater than the seasonal variability we observed. However, the two calibration sets are in a much better agreement in relative spectral terms. When the gains are normalized by the green channel gain, the ratios agree to about 3%. Further research is needed to account for these differences. One potential source of uncertainty is the solar spectral flux value used to convert the original ROLO radiance calibration factors to reflectance factor. Our future plans include deriving the EPIC calibration from Visible Infrared Imaging Radiometer Suite (VIIRS) data. This work may contribute to the resolution of the systematic difference.

**9 Conclusions**

We derived calibration coefficients for four EPIC visible and NIR channels by two independent methods using collocated MODIS Aqua and Terra scenes. The methods were applied to the initial (Version 1) and recent (Version 2) releases of the data The gain coefficients calculated using the regression method and the MODIS/EPIC ratios method agree to within between 0.1% and 1.4%, respectively.  The effects of the straylight corrections were tested using the two EPIC data versions and were shown to be consistent with expectations. In addition spectral correction for comparison of the close EPIC and MODIS channels was implemented for the Version 2 data. Overall the application of the straylight and spectral corrections result in the successive reductions of the regression offset values in all channels increasing the confidence in the consistency of the calibration coefficients derivation.

 Seasonal variability was estimated from over one year long data record.  It was found to be less than 1% with no discernible trend.

Using EPIC Moon observations we calculated calibration coefficients for the B-band (688 nm) and the A-band (764 nm) channels. We assumed that there is a small difference between moon reflectances separated by 10 nm in wavelength and the gain coefficients in the adjacent red (680nm) and NIR (780nm) were used for this purpose. The values are therefore consistent and may be recommended for use together with the MODIS-derived coefficients for the non-absorbing channels (Table 2).

The gain coefficients developed in this study were found to agree to about 10% with ones independently derived from EPIC moon views using ROLO moon observations. The agreement improves to about 3% if the relative spectral gains normalized by the green channel value.

## Acknowledgments

The DSCOVR EPIC datasets were acquired from the DSCOVR project science team. The Terra and Aqua MODIS L1b data were acquired from the Level-1 and Atmosphere Archive & Distribution System (LAADS) Distributed Active Archive Center (DAAC), located in the Goddard Space Flight Center in Greenbelt, Maryland (https://ladsweb.nascom.nasa.gov/). We are grateful to Thomas Stone for providing the ROLO-derived calibration coefficients for EPIC and for the help in preparing the EPIC data for the use with ROLO model. We would like to thank Matt Kowalewski and Marshall Sutton for help with EPIC Lunar observations. In addition, we would also like to thank Karin Blank for her help with EPIC image geolocation and Alexander Cede for his help with EPIC straylight correction.

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

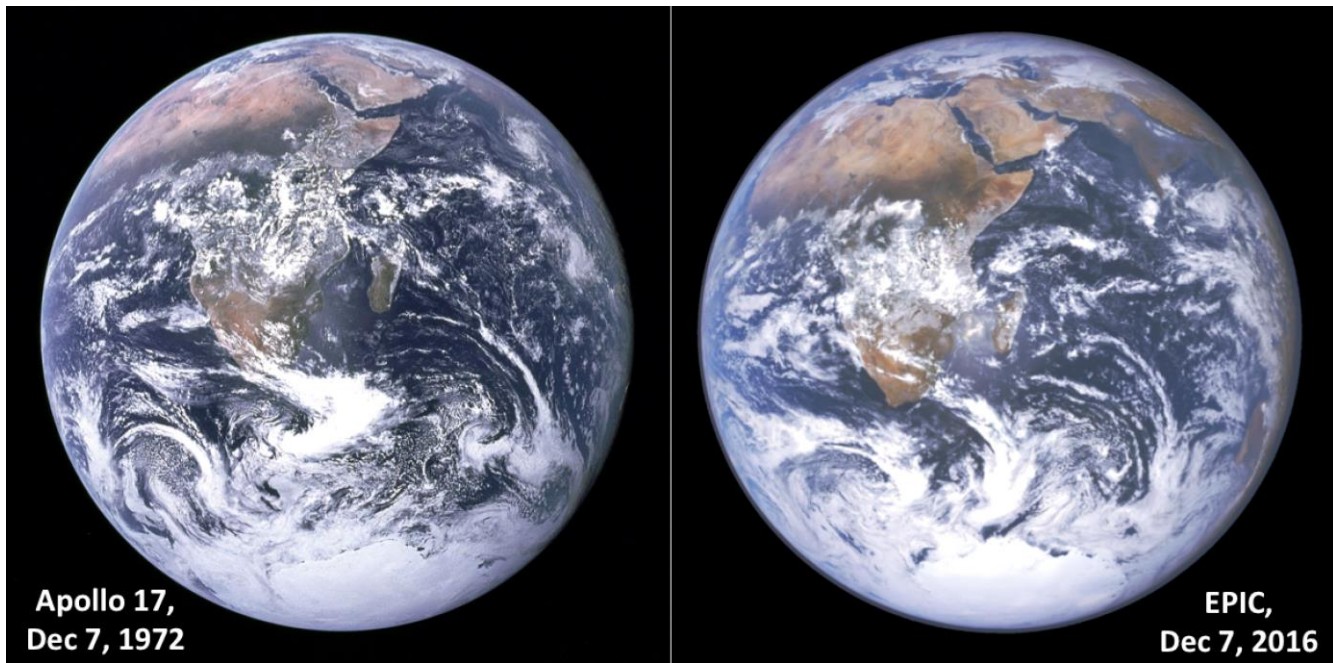

Figure 1: EPIC image (https://epic.gsfc.nasa.gov/?date=2016-12-07) of the entire sunlit Earth hemisphere (right) is compared with the Apollo (https://en.wikipedia.org/wiki/File:The_Earth_seen_from_Apollo_17.jpg) image taken on the same day 44 years ago. The two images show a remarkably similar large-scale cloud structure. (The EPIC image was recolorized to match the Apollo 17 image's Kodak Ektachrome film by Karin Blank, NASA/GSFC)

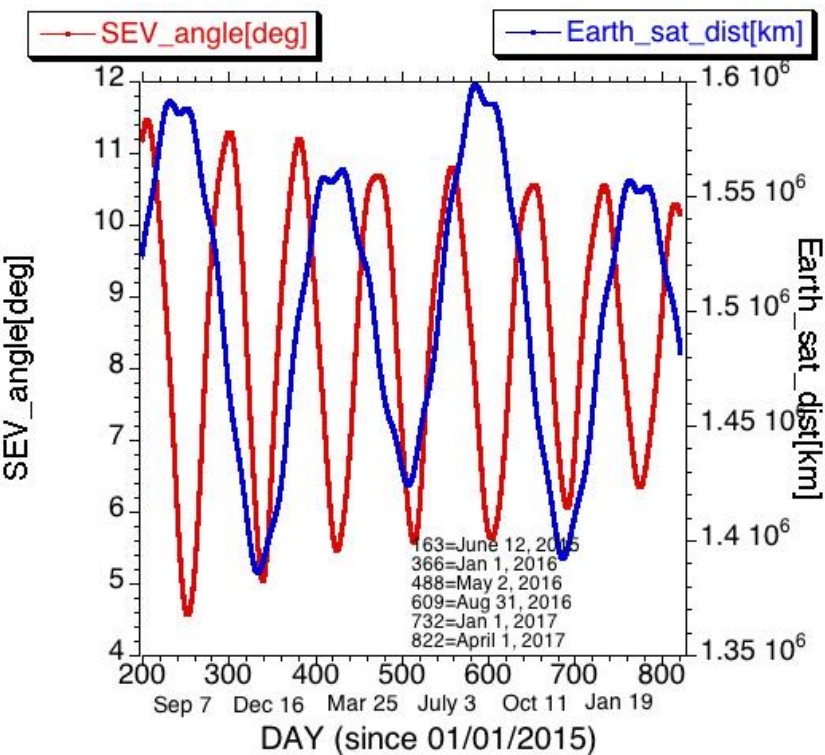

Figure 2: Solar Earth Vehicle (SEV) angle (left axis, red curve) and the distance between DSCOVR and Earth (right axis, blue curve) are plotted versus the day since January 1, 2015. Note that SEV = 180° – scattering angle between solar and viewing directions. Also note that the distance between DSCOVR and Earth changes approximately by 2000-2500 km per day.

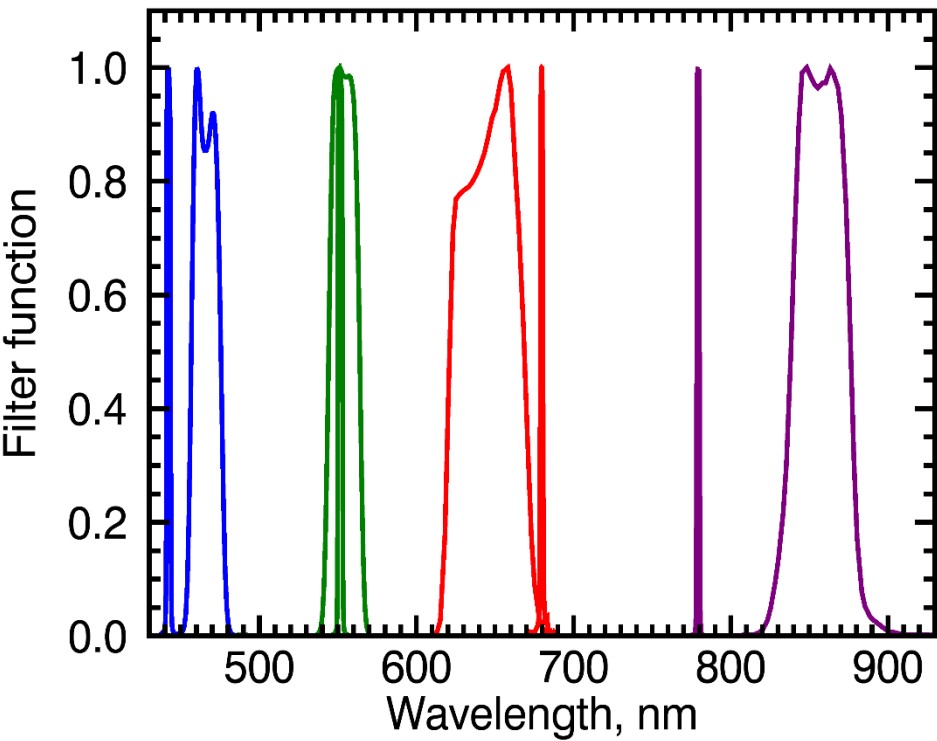

Figure 3: EPIC (narrow) and MODIS (wide) filter functions normalized to the maximum value for the four channels used in this study.

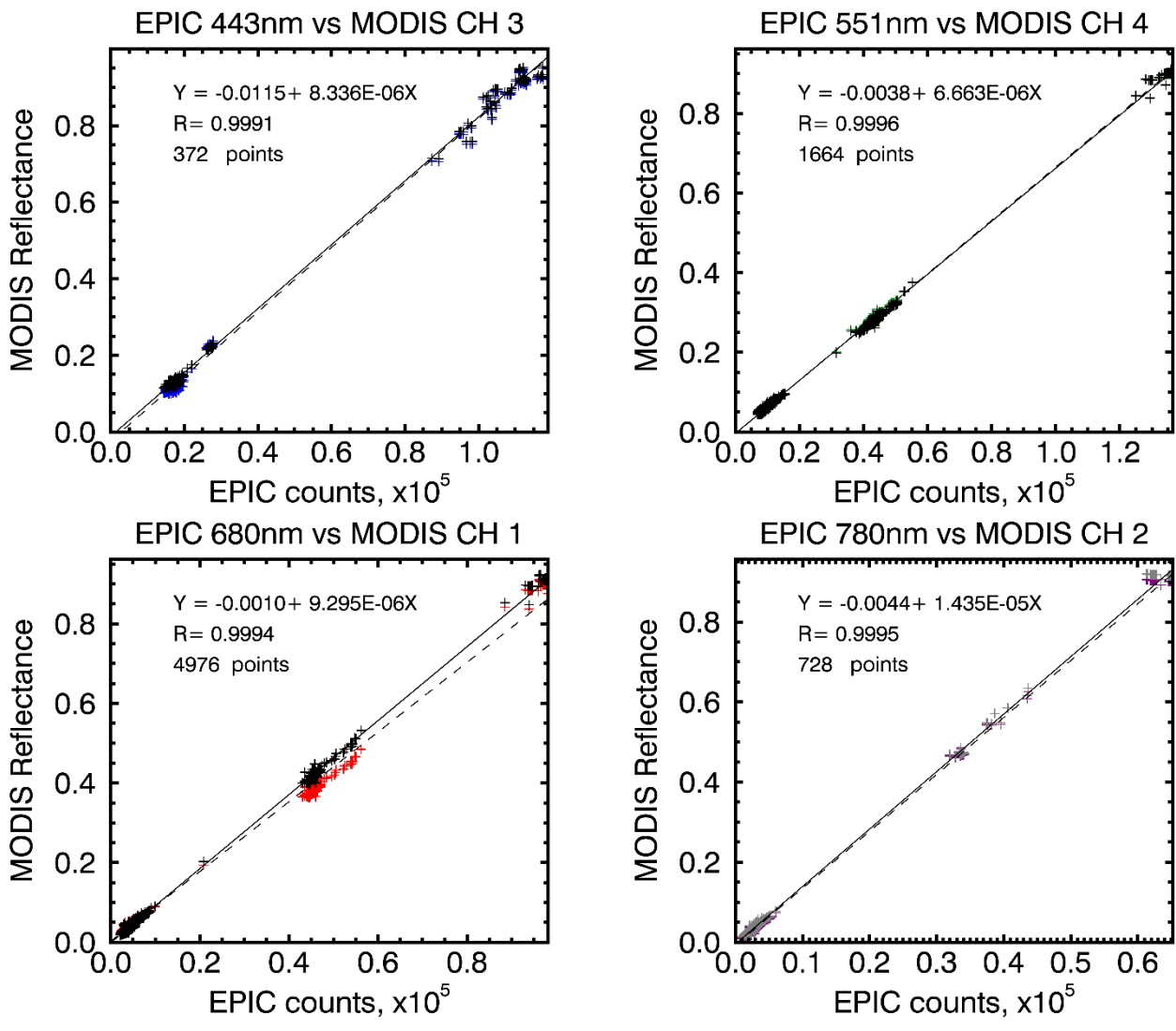

Figure 4: Scatter plots of the MODIS reflectance vs. EPIC Version 2 counts/sec in four spectral channels for the most homogeneous matching scenes and the corresponding regression lines and equations. The matches between June 2015 and March 2017 are used. Also shown are correlation coefficient R and the number of points used for the regression. The colored marks and dashed lines are for the data without spectral correction, the black marks and solid lines are for data with spectral correction.

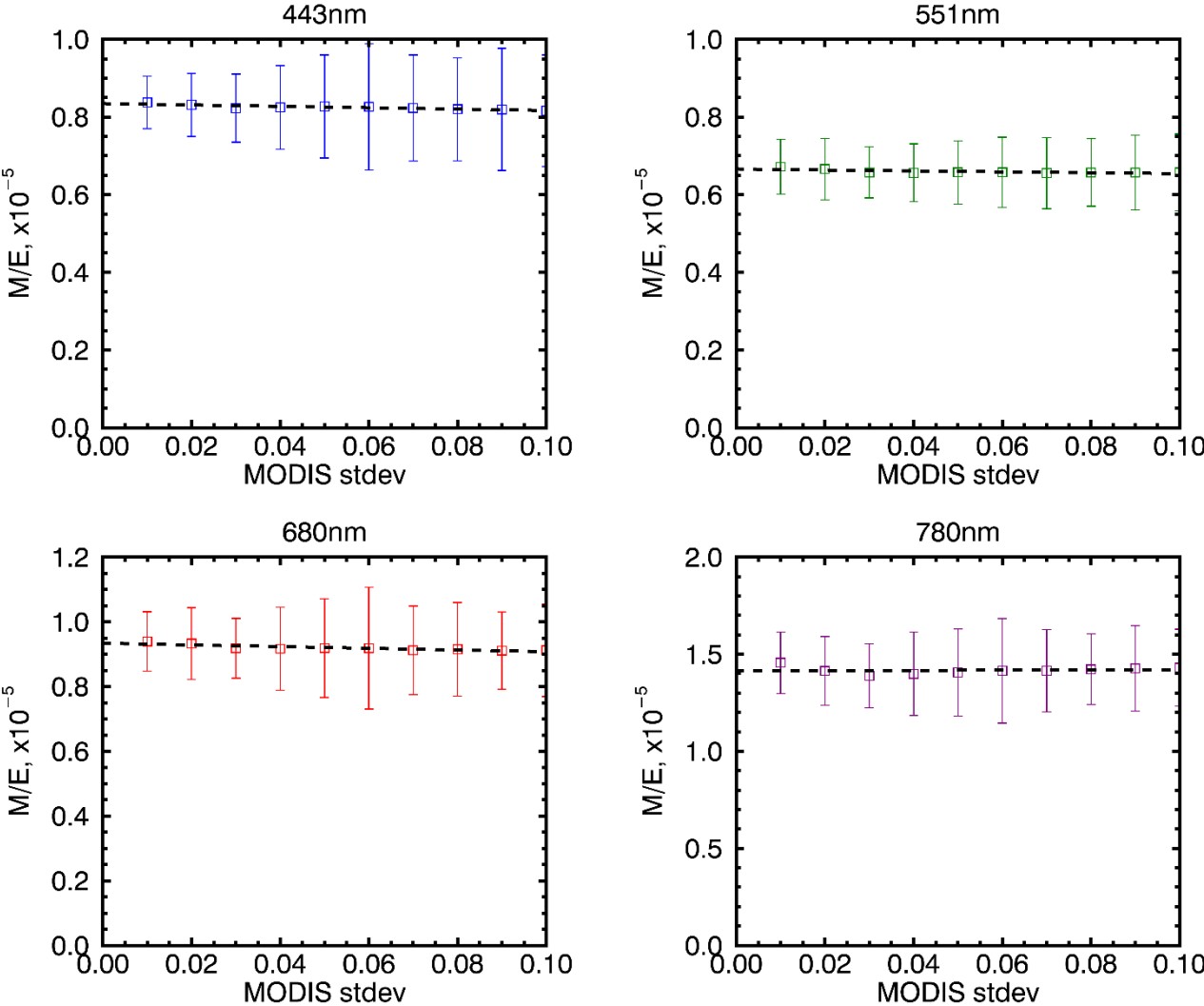

Figure 5: MODIS/EPIC ratios for the four spectral channels binned according to the MODIS relative standard deviation and MODIS reflectance > 0.6. Whiskers show mean and the standard variation of the ratios in each bin. Straight lines show linear regressions. EPIC Version 2 data. The resulting gain coefficients are 0.835 for the 443 nm channel, 0.666 for the 551 nm, 0.934 for the 680 nm and 1.41 for the 780 nm channel.

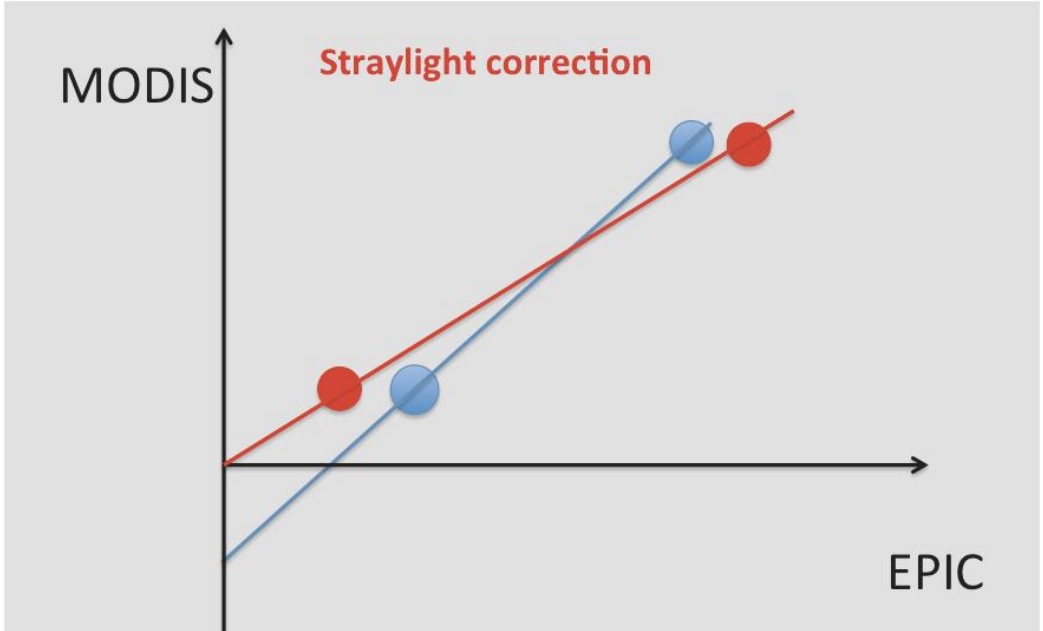

5    Figure 6: Schematic illustration of the effect of straylight correction. Blue dots and blue line represent a hypothetical regression fit for data without the straylight correction. Red dots and red line are for data with straylight correction. The correction decreases EPIC counts per second values for dark scenes and increases it for bright scenes, thus reducing the slope and the intercept of the fit. See the discussion in the text.

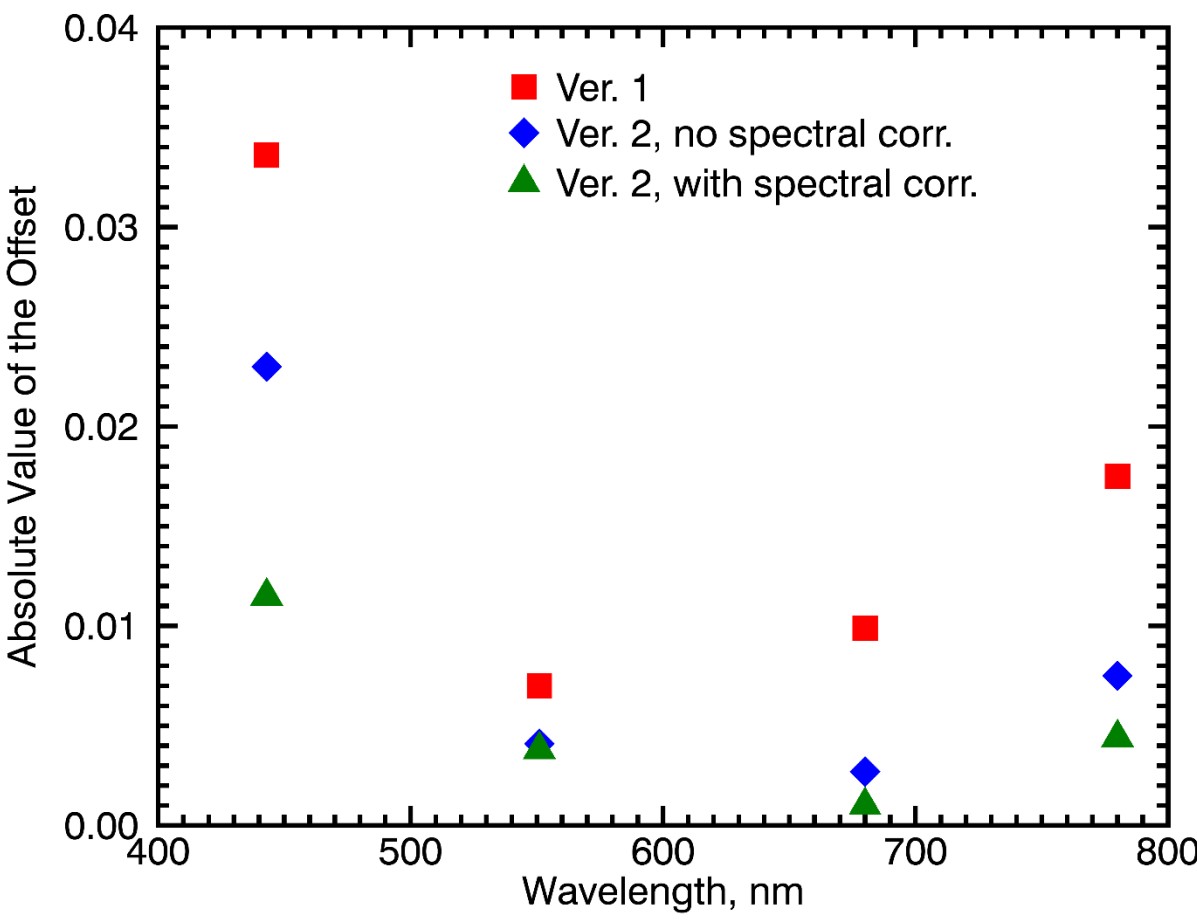

Figure 7: The absolute values of the regression offsets for Version 1 and Version 2 data with and without spectral correction. Note that the absolute values of offset are reduced due to both the straylight and the spectral corrections for all channels.

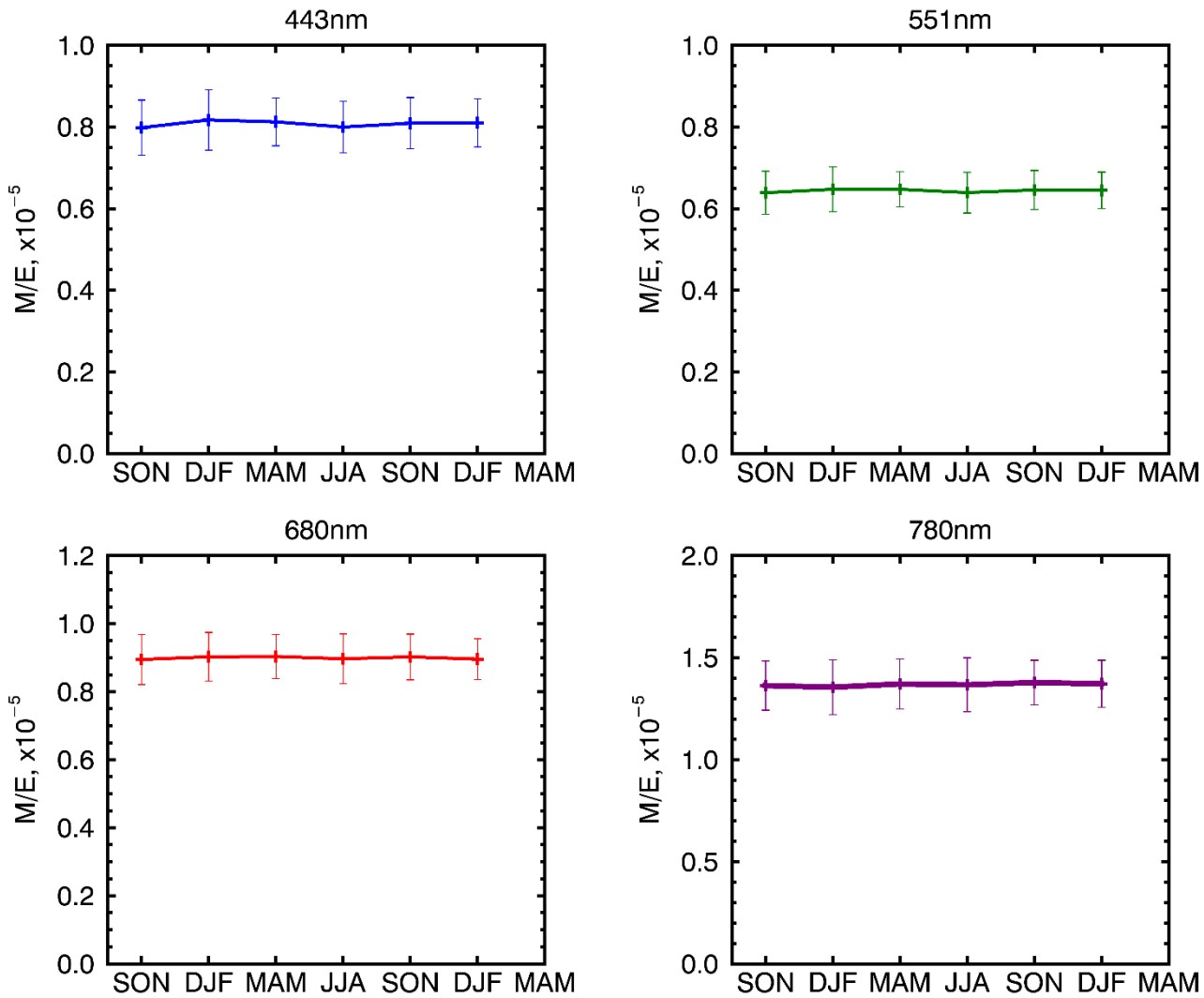

Figure 8: Seasonal dependence of the EPIC Version 2 calibration coefficients from September 2015 to February 2017. Whiskers represent the standard deviation of M/E ratios within each three-month period.

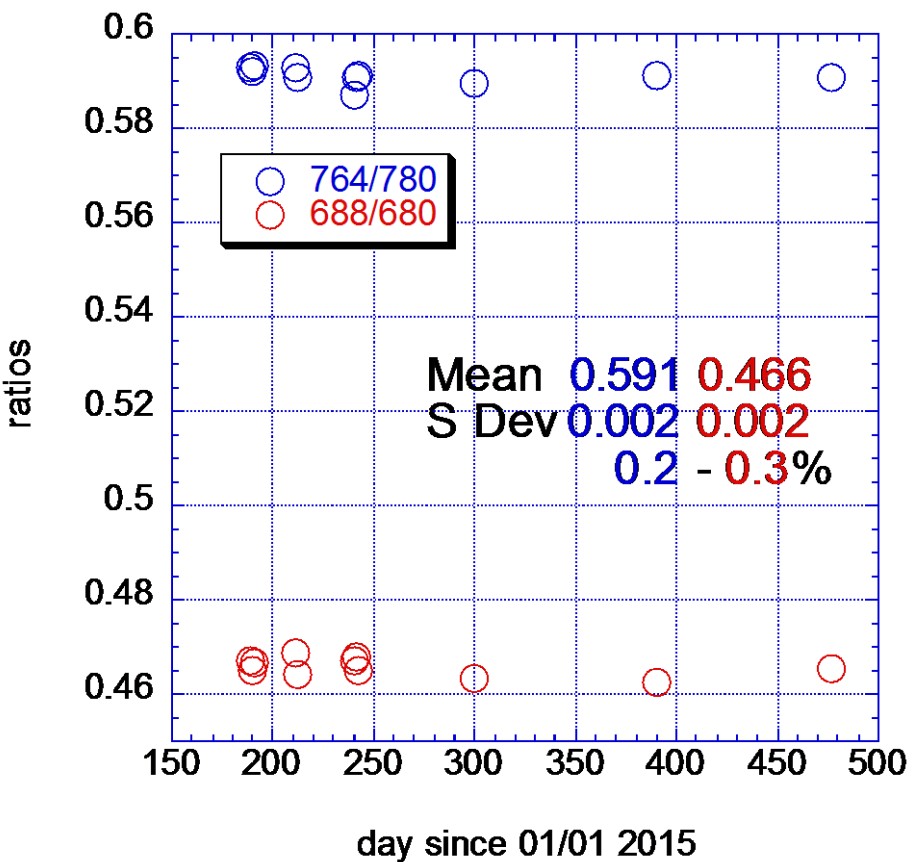

Figure 9: Ratios of O2 absorbing to reference channels using EPIC Moon observations. The data shown are for full moon. The new moon data on day 609 (not shown) provided the same channel ratios as the full moon data.

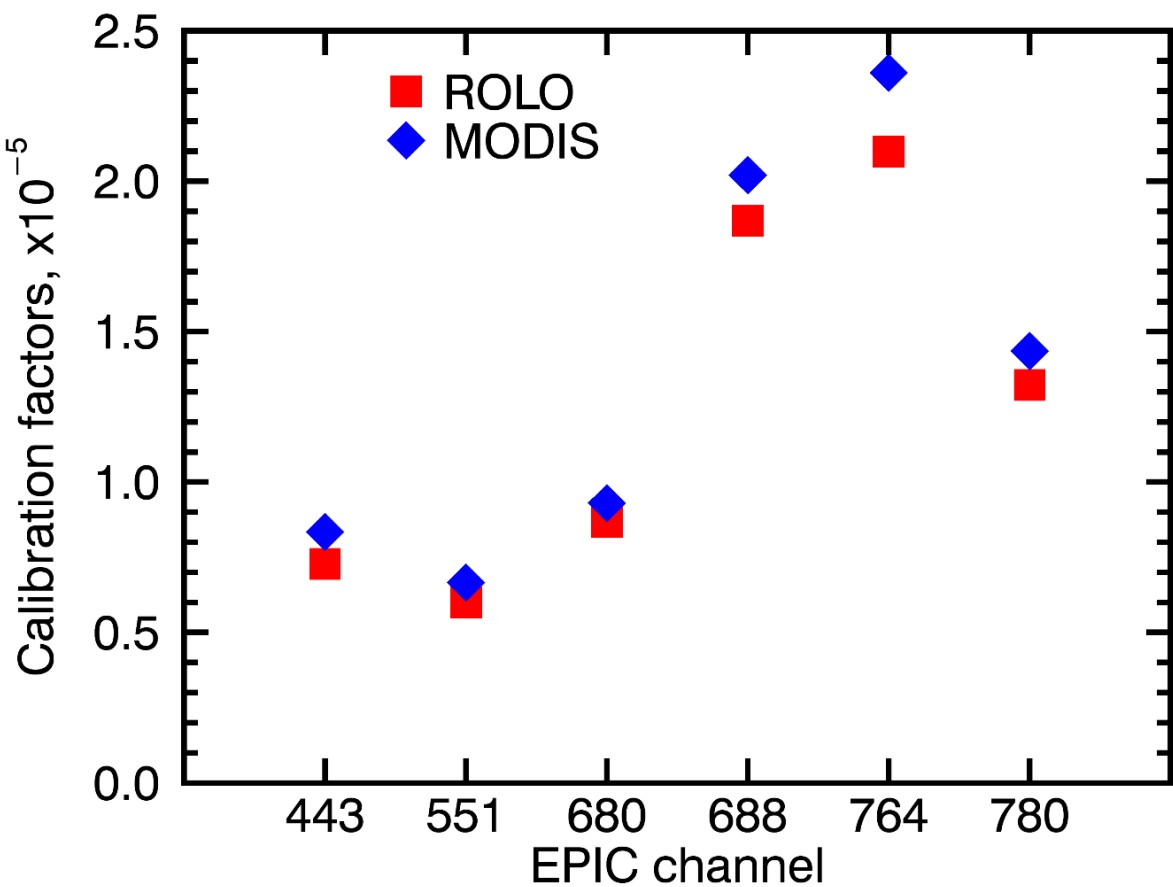

Figure 10: Comparison between MODIS- and ROLO-derived calibration coefficients.