# Peer review of "Calibration of the DSCOVR EPIC visible and NIR channels using MODIS Terra and Aqua data and EPIC lunar observations"

_Atmospheric Measurement Techniques, 2017_

## Referee Comment (RC1) · Anonymous Referee #2 · 2 Sep 2017

The paper presents two methods to calibrate DSCOVR visible and NIR channels. The first method uses MODIS reflectance vs. DSCOVR digital count regression, while the second method uses MODIS reflectance vs. digital count ratio as a function of MODIS reflectance standard deviation. The paper overall is sound, but to generate the community's excitement, it needs to add the unique sciences that are already published, not only by the authors, but also by others. Furthermore, the text, techniques, and figures/figure captions needs to be improved to increase clarity. The paper needs to address the following concerns before it be accepted.

0. there is little description about the scientific use of EPIC. Why do we need to calibrate EPIC in the first place? Has any interesting work done regarding the retrieval of aerosols, clouds, and surface properties? Any recent publications regarding the use of EPIC?

1. What is the radiometric resolution of MODIS vs. EPIC?

2. Do the spectral band adjustment factors consider the spectral response function difference between MODIS band and EPIC band? This is very important, as the reflectance depends on the spectral response function of each channel.

3. The results show ~10% difference with another independent method. There is little discussion about how to reconcile such difference? Are 10% difference small? How 10% or 3% differences may affect the level-2 products?

4. Some description about MODIS calibration and its accuracy should be discussed.

5. The figure captions should be sufficiently to readers to understood the figure. Figure 6. what are red dots, and what are blue dots? can an example with real data be shown here? Fig. 7. why use absolute values of regression offset? what is the difference between gain coefficient in fig. 8 vs. calibration coefficients in Fig. 10?

---

## Referee Comment (RC2) · Anonymous Referee #1 · 28 Sep 2017

Summary This study calibrates the EPIC imager channels with corresponding MODIS band calibration using two different methods, that were found to agree. The application of the SBAF and accounting for stray light show that the regression offsets are closer to the true instrument offset of zero in version 2 EPIC. The EPIC absorbing channels were calibrated using lunar targets after adjusting for a small spectral shift. The absorbing channel calibrations using this method were compared to ROLO and was found to be within 10%. This paper is ready for review after the following issues have been addressed.

I agree with the other reviewer. What is the mission of DSCOVR? Why is the calibration

needed? What is being retrieved from DSCOVR? Why are the channel spectra so narrow? Must be for trace gas retrieval, such as ozone. I can't believe that there are no DSCOVR publications that can be cited in this paper.

Page 1 line 30. Can you also provide the range of the scattering angle for a sun-synch satellite such as NPP as a reference? A large scattering angle increases the uncertainty of a retrieval, for example clouds, since it is nearly in direct backscatter. Can you state what retrieval would benefit from such a large scattering angle? Can you also state that a satellite in L1 would have to orbit L1 in order to be in the L1 orbit. Why is the range of SEV decreasing over time? Is the orbit about L1 maintained?

Page 2 line 16: The MODIS channel reflectances are not truly reflectances, that is dependent on the solar zenith, but a scaled radiance, that is divided by the solar constant of the channel. The reader needs to be aware of this in Fig. 4

Page 4 line 5: The pixel-level homogeneity threshold was set as a function of channel. Can the range of the spatial homogeneity threshold be given as a percentage of the mean pixel value? Was the spatial homogeneity threshold the greatest limiting factor of the number of EPIC and MODIS pairs?

Fig. 4: Can the authors identify the 3 groups of reflectance pairs. Is it clear-sky ocean, clear-sky Saharan desert, and bright clouds. Is the strict pixel-level standard deviation threshold, screening out more bright deep convective clouds or maritime stratus clouds? Each of these scene-types would require differing spectral band adjustment factors.

Table 2: Do the calibration coefficients that are published assume a zero offset? When comparing M/E ratio, does this represent calibration approach one with a zero-offset?

Table 2: can both approach 1 and 2 calibration coefficients also be added to Table 2.

Fig. 5: Why do you believe that there is a dependency of the EPIC gain with the MODIS/EPIC ratio standard deviation? In order to justify a linear regression based on

the ratio standard deviation to find the true ratio. Why do believe this is systematic rather than random?

Fig 5: Does the EPIC instrument angular configuration allow for sunglint? I guess since sun-glint is only a forward scatter feature, this would not be the case. Bright sunglint can also exceed 0.6 reflectance.

Page 6 line 10. How confident are the authors that the bright clouds are deep convective, rather than maritime stratus, which have differing SBAFs.

Page 6 line 18. Intermediate brightness scenes. Since there are so few EPIC MODIS reflectance pairs, could not the authors identify the actual scene. It is likely that these scenes are clear-sky deserts, since the deserts are more spectrally red than clouds, have a very different SBAF as shown in Fig. 4, than bright clouds.

Figure 7. Its good to see the SBAF correction changes the linear regression offset closer to the true space offset of 0.

Where are tables 3 and 4?

Table 5. Could the authors add to table 5, the actual EPIC gain factors from both methods and a recommendation of which EPIC calibration gains to use?

Page 7 line 1: are you trying to find seasonal dependence of the calibration method or EPIC sensor seasonal dependency. Can this be more clearly stated.

First of all do you expect any sensor degradation of EPIC? The sensor is at L1 where there is so little reflected solar exposure to the optics.

Evaluating a seasonal cycle, with one seasonal cycle is difficult. After 2-years than the actual seasonal dependence can be determined with more certainty. It is also interesting that the larger ratio disparities have a seasonal cycle.

Page 7 line 18. Regarding, the 0.688 and $0.680\mu$m lunar reflectance difference of 1.6%. Is that the bright portions of the moon or the dark portions? I guess what I

am asking do you use the complete lunar disc to get the ratio between the 0.688 and 0.680$\mu$m channels. Do you account for lunar phase and libration?

ROLO section. Did Tom Stone offer guidance to prepare the EPIC data to be compared with the ROLO model?

Conclusions, Page 9, line 8. Can you provide the EPIC calibration gains for the 0.764$\mu$m and 0.688$\mu$m here and some text why you recommend it?

---

## Author Comment (AC1) · 26 Oct 2017

We would like to thank the Anonymous Referee #2 for their insightful comments which helped us generate what we hope is a much improved manuscript. The paragraphs with the Referee's comments below start with ">" symbol, followed by our responses

> The paper presents two methods to calibrate DSCOVR visible and NIR channels. The first method uses MODIS reflectance vs. DSCOVR digital count regression, while the second method uses MODIS reflectance vs. digital count ratio as a function of MODIS reflectance standard deviation. The paper overall is sound, but to generate the community's excitement, it needs to add the unique sciences that are already published, not only by the authors, but also by others. Furthermore, the text, techniques, and figures/figure captions needs to be improved to increase clarity. The paper needs to address the following concerns before it be accepted.

We agree with the questions raised by the Reviewer and we made specific changes described below to address her/his concerns.

> 0. there is little description about the scientific use of EPIC. Why do we need to calibrate EPIC in the first place? Has any interesting work done regarding the retrieval of aerosols, clouds, and surface properties? Any recent publications regarding the use of EPIC?

We agree and have significantly extended the description of the scientific applications of the EPIC data and added multiple references describing these applications in detail. The end of the first paragraph in the introduction section was modified as follows: "Thanks to its position and viewing geometry, the EPIC instrument offers an improved temporal sampling compared to instruments on the sun-synchronous orbit. It samples the entire sunlit hemisphere 10-20 times per day. Compared to other instruments on geostationary orbit, EPIC provides improved coverage in high latitudes hemispheres. It thus has the potential to augment remote sensing observations in such applications as aerosol, cloud, sulphur dioxide and ozone amounts as well as vegetation properties (Marshak et al., 2017a). EPIC data are used for the remote sensing of height and optical depth of dust plumes using oxygen A and B bands (Xu et al., 2017, Yang et al., 2013) and multi-spectral UV SO2 measurements of the sunlit Earth disk (Carn et al., 2016). EPIC measurements are applied to the estimation of leaf area index and its sunlit portion (Yang et al., 2017; Marshak and Knyazikhin, 2017) as well as measuring the ozone, cloud reflectivity and erythemal irradiance (Herman et al., 2017). EPIC measurements were used to observe the terrestrial glint from oriented ice crystals by (Marshak et al., 2017b)."

> 1. What is the radiometric resolution of MODIS vs. EPIC?
The radiometric resolution of MODIS and EPIC instruments is 12 bit per pixel. We have included this information in the first and second paragraphs in the Data section.

> 2. Do the spectral band adjustment factors consider the spectral response function difference between MODIS band and EPIC band? This is very important, as the reflectance depends on the spectral response function of each channel.

Yes they do. To clarify this we have modified the third sentence of the first paragraph; it reads as follows: "These factors in the form of linear regression coefficients were obtained from https://cloudsgate2.larc.nasa.gov/cgi-bin/site/showdoc?mnemonic=SBAF; they are based on the analysis of the SCHIAMACHY hyperspectral data for various surface targets to account for the differences in MODIS and EPIC spectral response functions (Scarino et al., 2016)."

> 3. The results show 10% difference with another independent method. There is little discussion about how to reconcile such difference? Are 10% difference small? How 10% or 3% differences may affect the level-2 products?

To put the observed difference in context we modified the end of the last paragraph to read as follows: "The difference with the ROLO coefficients is noticeably greater than the two methods reported in the previous sections and greater than the seasonal variability we observed. However, the two calibration sets are in a much better agreement in relative spectral terms. When the gains are normalized by the green channel gain, the ratios agree to about 3%. Further research is needed to account for these differences. One potential source of uncertainty is the solar spectral flux value used to convert the original ROLO radiance calibration factors to reflectance factor. Our future plans include deriving the EPIC calibration from Visibly Infrared Imaging Radiometer Suite (VIIRS) data. This work may contribute to the resolution of the systematic difference."

We believe that widening the scope of the paper to include a discussion of the effects of calibration accuracy on the L2 EPIC-derived products would not be justified given
a significant number and disparate nature of such products (aerosol, cloud, sulphur dioxide and ozone amounts as well as vegetation properties). In addition, each product is developed by a different science team with a better knowledge of the subject. Finally, the EPIC Level 2 products have not been yet released. We plan to analyze the effect of Level 1 data uncertainties on Level 2 products when the Level 2 products will be available (the end of 2017).

> 4. Some description about MODIS calibration and its accuracy should be discussed.

We added a reference to Toller et al. (2013) that discusses the accuracy of MODIS level 1B calibration to the last paragraph of the Data section.

> 5. The figure captions should be sufficiently to readers to understood the figure. Figure 6. what are red dots, and what are blue dots? can an example with real data be shown here?

We have extended the Figure 6 caption to read as follows: Figure 6: Schematic illustration of the effect of straylight correction. Blue dots and blue line represent a hypothetical regression fit for data without the straylight correction. Red dots and red line are for data with straylight correction. The correction decreases EPIC counts per second values for dark scenes and increases it for bright scenes, thus reducing the slope and the intercept of the fit. See the discussion in the text. It is difficult to use the real data for visualization due to the small magnitude of the changes. We think a schematic representation better illustrates the effect.

> Fig. 7. why use absolute values of regression offset?

We added the following explanation to the last paragraph of the Spectral correction section: "The closeness of the offset values to the ideal case of zero offset can be interpreted as an improvement."

> what is the difference between gain coefficient in fig. 8 vs. calibration coefficients in Fig. 10?

AMTD
They are the same quantity. For consistency we now use "calibration coefficients" in both captions

---

## Author Comment (AC2) · 26 Oct 2017

We would like to thank the Anonymous Referee #1 for their insightful comments which helped us generate what we hope is a much improved manuscript. The paragraphs with the Referee's comments below start with ">" symbol, followed by our responses

>Summary. >This study calibrates the EPIC imager channels with corresponding MODIS band calibration using two different methods, that were found to agree. The application of the SBAF and accounting for stray light show that the regression offsets are closer to the true instrument offset of zero in version 2 EPIC. The EPIC absorbing channels were calibrated using lunar targets after adjusting for a small spectral shift.

[Figure]

The absorbing channel calibrations using this method were compared to ROLO and was found to be within 10%. This paper is ready for review after the following issues have been addressed.

>I agree with the other reviewer. What is the mission of DSCOVR? Why is the calibration needed? What is being retrieved from DSCOVR? Why are the channel spectra so narrow? Must be for trace gas retrieval, such as ozone. I can't believe that there are no DSCOVR publications that can be cited in this paper.

We agree with this comment.

As far as we know the EPIC channels' width is a part of its filter-wheel design.

We have significantly extended the description of the scientific applications of the EPIC data and added multiple references describing these applications in detail. The end of the first paragraph in the introduction section was modified as follows: "Thanks to its position and viewing geometry, the EPIC instrument offers an improved temporal sampling compared to instruments on the sun-synchronous orbit. It samples the entire sunlit hemisphere 10-20 times per day. Compared to other instruments on geostationary orbit, EPIC provides improved coverage in high latitudes hemispheres. It thus has the potential to augment remote sensing observations in such applications as aerosol, cloud, sulphur dioxide and ozone amounts as well as vegetation properties (Marshak et al., 2017a). EPIC data are used for the remote sensing of height and optical depth of dust plumes using oxygen A and B bands (Xu et al., 2017, Yang et al., 2013) and multi-spectral UV SO2 measurements of the sunlit Earth disk (Carn et al., 2016). EPIC measurements are applied to the estimation of leaf area index and its sunlit portion (Yang et al., 2017; Marshak and Knyazikhin, 2017) as well as measuring the ozone, cloud reflectivity and erythemal irradiance (Herman et al., 2017). EPIC measurements were used to observe the terrestrial glint from oriented ice crystals by (Marshak et al., 2017b)."

> Page 1 line 30. Can you also provide the range of the scattering angle for a sunsynch

satellite such as NPP as a reference?

We added the following explanation to the first paragraph in the Introduction section: "For comparison, depending on the season, latitude and scan view angle, the scattering angle for MODIS is typically in a wide range of between 110° and 175°. The Suomi-NPP VIIRS instrument, due to its wider range, covers even larger range of angle including the whole backscattering region."

> A large scattering angle increases the uncertainty of a retrieval, for example clouds, since it is nearly in direct backscatter. Can you state what retrieval would benefit from such a large scattering angle?

We agree that large scattering angles may present challenges for some retrievals. However they may also be desirable for other applications. We therefor added the following discussion to the top of page 2: "The almost back scattering EPIC observations are a direct consequence of its position at L1. The large scattering angle of EPIC observations is a significant difference compared to the observations from low orbit instruments. The large scattering angles may present challenges for some retrievals. However they may also be desirable for other applications. For example, the position of the water surface glint in the center of the sunlit hemisphere allows better coverage where LEO instruments often see glints as big in the Indian Ocean. Also of note is the lack of shadows for vertically extended scenes. Measurements in the backscattering region allowed to observe and characterize the glint caused by oriented ice crystals in clouds (Marshak et al. 2017b). Availability of these measurements also allow to better characterize BRDF of vegetation (Yang et al 2017; Marshak and Knyazikhin, 2017). Using the back scattering region it is possible to get Leaf Area Index of diffuse and sunlit leaves separately; this is important because they have different photosynthetic rates."

> Can you also state that a satellite in L1 would have to orbit L1 in order to be in the L1 orbit. Why is the range of SEV decreasing over time? Is the orbit about L1 maintained?

We added the following explanation to the first paragraph of the Introduction section: "The spacecraft actively maintains itself to be in a Lissajous orbit around L1."

According to the explanations we received from the mission control people the range of the variation of the Solar-Earth-Vehicle angle (SEV = 180o – scattering angle between solar and viewing directions) has been decreasing due to the evolution of its orbit since the launch but is expected to start widening again."

> Page 2 line 16: The MODIS channel reflectances are not truly reflectances, that is dependent on the solar zenith, but a scaled radiance, that is divided by the solar constant of the channel. The reader needs to be aware of this in Fig. 4

We agree and included the following clarification to the second paragraph in the Data Section: "Note that the MODIS reflectance, as well as EPIC, is the true reflectance multiplied by the solar zenith angle (MODIS Level 1B Product User's Guide, 2006). We will refer to this quantity as simply "reflectance".

> Page 4 line 5: The pixel-level homogeneity threshold was set as a function of channel. Can the range of the spatial homogeneity threshold be given as a percentage of the mean pixel value? Was the spatial homogeneity threshold the greatest limiting factor of the number of EPIC and MODIS pairs?

Yes. We modified the end of the second paragraph of the Analysis section as follows: "The relative standard deviation of MODIS and EPIC points included in the regressions was between 0.5% and 1% depending on the channel. In this approach the spatial homogeneity threshold was the greatest limiting factor to the number of EPIC and MODIS pairs."

> Fig. 4: Can the authors identify the 3 groups of reflectance pairs. Is it clear-sky ocean, clear-sky Saharan desert, and bright clouds. Is the strict pixel-level standard deviation threshold, screening out more bright deep convective clouds or maritime stratus clouds? Each of these scene-types would require differing spectral band adjustment

factors.

"One may assume that bright pixels mostly represent cloudy scenes, dark pixels are mostly from water and vegetation, and intermediate values represent deserts. However, it would be impossible to classify the pixels with certainty within the framework of the crude approach, where SBAFS for clouds are assumed for bright scenes with reflectance greater 0.6 and a fixed surface classification map is used. This represents a limitation of the current approach and may be responsible for some spread in the regressions." We added the above comment to the first paragraph of the Spectral correction section. Please also see our response (*)

> Table 2: Do the calibration coefficients that are published assume a zero offset? When comparing M/E ratio, does this represent calibration approach one with a zero-offset?

Using the officially published gain coefficients implicitly assumes a zero offset. We provided offset values to each retrieval team. As far as we know, they were used by the aerosol retrieval team.

We included the following explanation before the last sentence in the Analysis Section: "The differences in the gain coefficients calculated using the two methods given in Table 2 which shows the officially published gain coefficients for the two dataset versions. These coefficients are also available at https://eosweb.larc.nasa.gov/project/dscovr/DSCOVR_EPIC_Calibration_Factors_V02.pdf. When comparing the gain coefficients from the two methods below we do not force the regression through zero. Doing so would reduce the independence of the two approaches as it would effectively ignore the contribution of the dark scenes to the regression."

> Table 2: can both approach 1 and 2 calibration coefficients also be added to Table 2.

We agree to the request in principle and the Version 2 calibration coefficients for both

methods are now available elsewhere in the paper (please see our response (\*\*)). The calibration coefficients for the Version 1 have only limited usefulness since the improved Version 2 data is out and new data is processed by the Version 2 algorithm. We believe that Table 2 provides the official calibration coefficients and adding more values to it may be confusing to the user.

> Fig. 5: Why do you believe that there is a dependency of the EPIC gain with the MODIS/EPIC ratio standard deviation? In order to justify a linear regression based on the ratio standard deviation to find the true ratio. Why do believe this is systematic rather than random?

We added the following discussion to the end of the Analysis Section. A dependence of the M/E ratio on the relative standard deviation of the MODIS pixels may potentially exist because of the different effect of the scene's cloud or surface inhomogeneity on the two instruments due to the different viewing geometry. However, this approach does not assume its existence, as the gain coefficients are obtained from the extrapolation to the "ideal" case of completely uniform scene thus accounting for any potential systematic behavior. If the relation to the standard deviation is completely random the resulting coefficients will be similar to what one would obtain by simply calculating the mean M/E ratio.

> Fig 5: Does the EPIC instrument angular configuration allow for sunglint? I guess since sun-glint is only a forward scatter feature, this would not be the case. Bright sunglint can also exceed 0.6 reflectance.

We agree and added the following sentence to the paragraph before the last one in the Analysis section: "Because EPIC observations are made in the backscattering region the sunglint usually occurs in the center of the image. Bright sunglint can exceed 0.6 reflectance however such scenes are not spatially homogeneous and are screened out by the relative standard deviation requirement"

> Page 6 line 10. How confident are the authors that the bright clouds are deep convective, rather than maritime stratus, which have differing SBAFs.

(*) We agree that it may not be possible to distinguish between the deep convective and maritime stratus clouds using the simple brightness threshold. The cloud identification problem presents itself for the SBAF derivation as well (Scarino et al., 2016). To evaluate the possible effect we rerun the fitting procedure using the "Approximate DCC" SBAF values from Scarino et al. (2016) which represent a collection of bright tropospheric clouds instead the "Precise DCC" classification, used in this paper. We found that the resulting difference in the calibration coefficients was on the order of 0.1% - significantly less than other sources of uncertainty.

> Page 6 line 18. Intermediate brightness scenes. Since there are so few EPIC MODIS reflectance pairs, could not the authors identify the actual scene. It is likely that these scenes are clear-sky deserts, since the deserts are more spectrally red than clouds, have a very different SBAF as shown in Fig. 4, than bright clouds.

We agree that the intermediate brightness scenes are likely to be deserts. One can not completely exclude partial cloud contamination however. We believe this is a limitation of the current approach and a source of uncertainty. Please also see our response to the comment about Figure 4 above and the resulting paper modifications.

> Figure 7. Its good to see the SBAF correction changes the linear regression offset closer to the true space offset of 0.

We agree

> Where are tables 3 and 4?

We have corrected this typo. Table 5 was renamed to Table 3.

> Table 5. Could the authors add to table 5, the actual EPIC gain factors from both methods and a recommendation of which EPIC calibration gains to use?

(**) We agree. The recommended (published) gain coefficients for EPIC version 1 and

2 data are given in Table 2. We now include the gain factors from both methods. For the regression method they are shown on Figure 4. For the M/E ratio method they are now included in the caption of Figure 5.

> Page 7 line 1: are you trying to find seasonal dependence of the calibration method or EPIC sensor seasonal dependency. Can this be more clearly stated. First of all do you expect any sensor degradation of EPIC? The sensor is at L1 where there is so little reflected solar exposure to the optics. Evaluating a seasonal cycle, with one seasonal cycle is difficult. After 2-years than the actual seasonal dependence can be determined with more certainty. It is also interesting that the larger ratio disparities have a seasonal cycle.

We agree with this comment and we modified the beginning of the "Seasonal dependence" as follows: "The length of the available EPIC dataset allowed us to evaluate the magnitude of any possible temporal change in the derived calibration coefficients. Such a change may potentially be due to two distinct factors: seasonal dependence of the calibration method itself or the degradation of the EPIC instrument. With the data covering only one full seasonal cycle it may be difficult to reveal a seasonal dependence of the calibration procedure and thus separate the two factors. However observing no or small temporal change would be an encouraging sign of both the stability of the instrument and the robustness of the calibration method."

> Page 7 line 18. Regarding, the 0.688 and 0.680_m lunar reflectance difference of 1.6%. Is that the bright portions of the moon or the dark portions? I guess what I am asking do you use the complete lunar disc to get the ratio between the 0.688 and 0.680_m channels. Do you account for lunar phase and libration? ROLO section. Did Tom Stone offer guidance to prepare the EPIC data to be compared with the ROLO model?

We added the following explanation to the section 7". To calculate the ratios we used both full moon and new moon data separately. To avoid the effects of libration the

edges of the disk were ignored."

Figure 1 shows that the F(680,688) and F (780,764) ratios for moon observations agree to within 1% and 2% respectively. The smaller size diamonds represent better data fit. It also shows that the new moon observation (circled) agrees well with the rest of the data.

Figue 2 shows the removal of the edge pixels of the moon image (in red, right panel) and examples of the EPIC moon views (left panels)

We extended the following acknowledgement to read: "We are grateful to Tomas Stone for providing the ROLO-derived calibration coefficients for EPIC and for the help in preparing the EPIC data for the use with ROLO model."

> Conclusions, Page 9, line 8. Can you provide the EPIC calibration gains for the 0.764_m and 0.688_m here and some text why you recommend it?

We added a reference to the values reported in Section 7 and included the following sentence: "The values are therefore consistent and may be recommended for use together with the MODIS-derived coefficients for the non-absorbing channels (Table 2).
* * *
**Lunar calibration of the O2 abs. channels**

$F(680,688)$

and

$F(780,764)$

[Figure]

*Courtesy of Marshall Sutton*

17 images without the stray light cor. and 36 images with the stray light cor.

The difference in 764/780 ratio < 2%, and the difference in the 688/680 ratio < 1%.

**Fig. 1.**

[Figure]

EPIC R05-V02 Lunar Images
SciGlob, 2017-05-05

R05-V02: 09/01/2016 - 680nm

Examples of moon observations in 680 (left) and 688 (right) on 09/01/2016 (courtesy of Matthew Kowalewski).

**Fig. 2.**

---

## Author Response (AR2)

**Response to the Anonymous Referee #1**

Summary
This study calibrates the EPIC imager channels with corresponding MODIS band calibration using two different methods. They apply SBAF and show that the regression offsets are closer to a zero offset in version 2 EPIC, which has accounted for stray light. The EPIC absorbing channels were calibrated using lunar targets after adjusting for a small spectral shift. The absorbing channel calibration using this method was compared to ROLO and was found to be within 10%. This paper is ready for review after the following issues have been addressed.

I appreciate the authors adding a section of how the EPIC observations can be used scientifically.
I have few more minor comments. After these have been addressed I approve the paper for publication. I do not need to review the revised paper.

We are grateful for the Referee's detailed and insightful comments. Below we provide our responses.

Specific Comments

Page 2 line 8. glints-> glint

We have corrected the typo.

Page 3 line 12. EPIC has 12-bit radiometric resolution. It samples the Earth 10-20 times a day. It combines 4 pixels (8^2 km) into 1 pixel (18^2 km). It seems odd, to lower the spatial resolution, rather, than lower the radiometric bit rate from 12 to 10 bits, or to increase the number of Earth images per day. Many geostationary instruments have only 10-bit resolution. Can the authors explain why keeping a 12-bit resolution is so important for the EPIC retrievals?

We share the referee's curiosity. While we do not feel sufficiently qualified to discuss specific spacecraft design decisions in the paper, we understand that a number of factors, not necessarily of purely scientific nature, may have contributed such as the amount of on-board storage and the availability of the deep-space communication antennae.

Page 3 line 21. For a sensor at L1 it seems quite odd to make the spectral response functions extremely narrow, especially if you are trying to capture as much light as possible in order to increase the signal to noise ratio. The MODIS sensor is capable of retrieving clouds, aerosols, land use properties, with broad spectral bands. What retrievals need such narrow spectral response functions?

Similarly to the previous comment we do not feel sufficiently qualified to provide a definitive answer. From our discussions with people involved in EPIC design, we understand that the spectral width of EPIC channels is not linked to the gases, clouds or aerosol retrieval requirements but rather to its filter-wheel design. Specifically, please see "Rotating shutter and exposure times" section at https://epic.gsfc.nasa.gov/epic which states that "In the refurbished design, the filter widths were adjusted ... to improve the uniformity of exposure across the CCD."

Page 3 line 27. Is there a reference that describes the Version 2 improvements?

We are not aware of any publications that detail the differences between Version 1 and Version 2. The version 2 improvements have been briefly described in the manuscript titled "Earth Observations from DSCOVR/EPIC Instrument" that has been recently submitted to BAMS" We added this reference to the text of the paper.

Page 4 line 7. What is the dwell time to make an observation? How much does the Earth rotate underneath during the measurement time?

We added the following explanations:
The exposure time varies between 0.046 second for the 443nm channel to 0.025 second for the 780 nm channel. During these times the Earth rotates less than 20 meters. Compared to many low-orbit radiometers that perform cross-track scans to create an image, EPIC filter-wheel design means that image acquisition is nearly instantaneous. However, there is a delay of ~3 min between blue (443 nm) and green (551 nm) and ~4 min between blue and red (680 nm). In 4 min a point on the equator will rotate approximately 110 kilometers. The geolocation algorithm ensures the spatial collocation of the spectral channels for each pixel.

Figure 4. Are these all the EPIC and MODIS matches from June 2015 and March 2017? Please state this in the figure caption.

We added the following sentence to the caption: "The matches between June 2015 and March 2017 are shown."

Page 4 line 28. The pixel homogeneity was the greatest limiting factor. Less than 5000 matches were found, during a time period of 1 ¾ years. The pixel homogeneity of both MODIS and EPIC must be small. This is one way to avoid having to deal with navigation differences.

Has the version 2 navigation improved over version 1? What makes the navigating this sensor so difficult?

The accuracy of the geolocation did indeed improved in Version 2. The much larger distance between the Earth and the spacecraft compared to the low earth orbit instruments and strong effects of the Earth's sphericity complicate EPIC geolocation. We added the following to the end of section 4: "An improved agreement between the two methods for Version 2 data may partially be attributed to more accurate geolocation algorithm. Other factors such as the straylight correction and spectral correction are discussed in the following sections."

In the acknowledgements, you state that Karen Blank helped out with the EPIC image geolocation. Is this an improvement to the version 2 navigation that she implemented? To make the EPIC data more useful, how was the EPIC data for this study navigated improved upon, which was found in the version 2 data?

Karin Blank is the person responsible for the EPIC geolocation algorithms for both version 1 and 2 EPIC data. She developed and implemented a number of improvements to the geolocation algorithm. It is briefly described in the manuscript titled *"Earth Observations from DSCOVR/EPIC Instrument"* recently submitted to BAMS.

Page 5 line 4. It is fortunate that the EPIC sensor is not degrading over time. If it were degrading the first approach could not be used to monitor the degradation. Also, if one picks a reflectance threshold using the same channel that needs to be calibrated, and it is degrading, the true sensor degradation would be underestimated. Would it not be better approach to use a frequency threshold, as you suggested as 10% or 15%?

The threshold requirement is only applied to the matching MODIS pixels which are assumed to be well calibrated and stable. The only requirement for the matching EPIC pixels is the homogeneity of the neighborhood (small relative standard deviation). We added this clarification to the text.

Page 5 line 12. But the first approach is crucial since it verifies that the sensor response is linear. However, the lack of sampling over the dynamic range, the linearity of the sensor cannot be verified, using approach one in its current form. But, I do agree that approach one does show that the stray light corrections are causing the linear regressions offset to be closer to zero.

We agree.

Figure 5. The 780nm M/E ratio in the caption of 1.141 does not match the M/E in the plot of ~1.4.

We have corrected this typo.

Page 7 line 10. SCHIAMACHY -> SCIAMACHY

Corrected.

Page 9 line 13. Did the full moon and new moon data provide the same channel ratios? Which ratios are provided on line 15? Can the lunar phase symbols be used rather open circles to assure the readers that the ratio is the same for new and full moon conditions?

The data in Fig. 9 are all for full moon.  The new moon data were used on day 609 not shown in Fig. 9.  The new moon data provided the same channel ratios as the full moon data.
We added the above information to the figure caption.